# K-RAS Associated Gene-Mutation-Based Algorithm for Prediction of Treatment Response of Patients with Subtypes of Breast Cancer and Especially Triple-Negative Cancer

**DOI:** 10.3390/cancers14215322

**Published:** 2022-10-28

**Authors:** Heather Johnson, Amjad Ali, Xuhui Zhang, Tianyan Wang, Athanasios Simoulis, Anette Gjörloff Wingren, Jenny L. Persson

**Affiliations:** 1Olympia Diagnostics, Inc., Sunnyvale, CA 94086, USA; 2Department of Molecular Biology, Umeå University, SE-901 87 Umeå, Sweden; 3Department of Bio-Diagnosis, Institute of Basic Medical Sciences, Beijing 100005, China; 4Department of Clinical Pathology and Cytology, Skåne University Hospital, SE-205 02 Malmö, Sweden; 5Department of Biomedical Sciences, Malmö University, SE-206 06 Malmö, Sweden

**Keywords:** machine learning algorithm, KRAS, breast cancer biomarkers, gene mutations, triple-negative breast cancer, luminal a breast cancer, progression-free survival, treatment response

## Abstract

**Simple Summary:**

Despite advances in treatment of subtypes of breast cancer, there still lacks reliable biomarkers with precision to predict treatment response at diagnosis. We used machine-learning tools and developed and validated a novel 12-Gene Algorithm as a biomarker for prediction of treatment response for breast cancer patients, especially those suffering triple-negative cancer. The 12-Gene Algorithm based on KRAS-associated gene-mutation profiles showed high accuracy at predicting the response of breast cancer patients including triple-negative subtype to first-line chemotherapy treatment in two independent patient cohorts. Our study suggests that the 12-Gene Algorithm has a potential to be used in clinical practice to improve breast cancer treatment decision-making, especially for triple-negative breast cancer patients.

**Abstract:**

Purpose: There is an urgent need for developing new biomarker tools to accurately predict treatment response of breast cancer, especially the deadly triple-negative breast cancer. We aimed to develop gene-mutation-based machine learning (ML) algorithms as biomarker classifiers to predict treatment response of first-line chemotherapy with high precision. Methods: Random Forest ML was applied to screen the algorithms of various combinations of gene mutation profiles of primary tumors at diagnosis using a TCGA Cohort (*n* = 399) with up to 150 months follow-up as a training set and validated in a MSK Cohort (*n* = 807) with up to 220 months follow-up. Subtypes of breast cancer including triple-negative and luminal A (ER+, PR+ and HER2−) were also assessed. The predictive performance of the candidate algorithms as classifiers was further assessed using logistic regression, Kaplan–Meier progression-free survival (PFS) plot, and univariate/multivariate Cox proportional hazard regression analyses. Results: A novel algorithm termed the 12-Gene Algorithm based on mutation profiles of *KRAS*, *PIK3CA*, *MAP3K1*, *MAP2K4*, *PTEN*, *TP53*, *CDH1*, *GATA3*, *KMT2C*, *ARID1A*, *RunX1*, and *ESR1*, was identified. The performance of this algorithm to distinguish non-progressed (responder) vs. progressed (non-responder) to treatment in the TCGA Cohort as determined using AUC was 0.96 (95% CI 0.94–0.98). It predicted progression-free survival (PFS) with hazard ratio (HR) of 21.6 (95% CI 11.3–41.5) (*p* < 0.0001) in all patients. The algorithm predicted PFS in the triple-negative subgroup with HR of 19.3 (95% CI 3.7–101.3) (*n* = 42, *p* = 0.000). The 12-Gene Algorithm was validated in the MSK Cohort with a similar AUC of 0.97 (95% CI 0.96–0.98) to distinguish responder vs. non-responder patients, and had a HR of 18.6 (95% CI 4.4–79.2) to predict PFS in the triple-negative subgroup (*n* = 75, *p* < 0.0001). Conclusions: The novel 12-Gene algorithm based on multitude gene-mutation profiles identified through ML has a potential to predict breast cancer treatment response to therapies, especially in triple-negative subgroups patients, which may assist personalized therapies and reduce mortality.

## 1. Introduction

The molecular subtypes of breast cancer (BC) are defined based on the presence or absence of expression of the steroid hormone receptors: estrogen receptor (ER) and progesterone receptor (PR) along with the growth factor receptor 2 (HER2). These major subtypes including (i) epidermal luminal A (ER+, PR+/− and HER2−), (ii) Luminal B (ER+, PR+/− and HER2+), (iii) HER2 (ER−/PR− and HER2+), and (iv) Triple-negative (ER−, PR−, HER2−) or basal-like breast cancers. The BC subtyping is important for cancer prognosis and neoadjuvant treatment decision-making [1,2,3,4]. Currently, no targeted therapy is available for triple-negative subtype of BC that lacks expression of ER, PR and HER2. Further, triple-negative tumors are of high complexity and heterogeneity, which respond poorly to endocrine therapies [5,6]. Moreover, triple-negative BC is profoundly associated with metastatic potentials and disease recurrence, thus it represents a major clinical challenge.

Despite advances in the treatment of ER+ and HER2+ subtypes of BC in the past decades, there is still a lack of reliable biomarkers with better precision than the existing ones currently used to predict treatment response of subtypes of BC, especially for triple-negative BC. The current biomarkers include clinicopathological factors/parameters, such as tumor size, histological grade, Ki67 expression, and expression of ER, PR and HER2-based immunohistochemistry (IHC) assays for BC subtyping [7]. Recently, the stromal tumor-infiltrating lymphocytes has been used as predictive biomarkers for risk stratification and treatment response [8,9]. However, the high variability in the pathological assessment limits the clinical accuracy of these biomarkers, and the errors may be inevitable. Moreover, due to the tumor heterogeneity in individual patients, the clinicopathological parameters are not robust enough to predict treatment response of BC. 

It is important to take into consideration that primary breast tumors harbor gain of function mutations in multiple oncogenes, while possessing loss of function mutations in multiple tumor suppressors. These oncogenes and tumor suppressors play important roles in cancer cell proliferation, survival and invasion in the presence and/or absence of therapeutic agents. Genome-wide screenings of large amounts of primary breast tumors have led to the identification of the co-existing of mutations in multiple genes including *KRAS, PIK3CA, MAP3K1, MAP2K4, PTEN* and *TP53* in primary breast tumors prior to treatment [10,11]. *KRAS* mutations with predominantly G12 codon mutations occur in 4% of approximately 500 primary breast tumors examined [12]. MDA-231 is a common triple-negative and metastatic BC cell line that harbors G-12D mutation [13]. *PIK3CA* mutations and *PTEN* loss are frequently observed in ER+ luminal subtype and triple-negative subtype of BC [5,14]. Although little is known about the precise role of *KRAS* mutations in progression of BC, it is well-established that PI3K/AKT/PTEN pathway is under control of KRAS activity and that abnormal activation of this oncogenic pathway promotes breast tumor growth and invasion [15,16]. In addition, KRAS is a major component of the mitogenic growth factor-dependent pathway, which is associated with epidermal growth factor receptor (EGFR), RAF and mitogen-activated protein-kinase (MAPK) pathways. EGFR treatment-resistance is mediated by mutations in *KRAS,* which lead to constitutive activation of the KRAS-MAPK pathway [17]. It has been shown that activation of KRAS-MAPK cascade can lead to the phosphorylation of ERα, which will in turn activate the transcriptional activity of the target genes of ERα and enable BC cells to gain survival advantages and bypass tamoxifen therapy-induced inhibition [18]. This suggests that KRAS-associated pathways influence response of BC cells to endocrine therapies by interfering with ERα activity. *KRAS* mutations may play an important role in rendering HER2 subtype of BC to become resistant to therapies [4,5]. It has been shown that co-existence of mutations in genes encoding the key transcriptional factors including *CDH1, GATA3, KMT2C, ARID1A* and *RunX1* are common in primary breast tumors, and these genes play central roles in cell cycle, survival, metabolism, motility, and genomic instability [19]. The combined mutational hotspots in members of the mitogen-activated protein kinase (MAPK) signaling cascades are frequently present in the HER2+ and triple-negative metastatic BC, which include *CDH1* mutations, *FOXA1* mutations [20] and *ESR1* mutations [21]. Mutations in the ligand-binding domain of *ESR1* were found in 18% of endocrine-resistant HER2+ BC. This suggests that the development of predictive models for BC diagnosis and prognosis needs to take into consideration of cancer genomic data that reflects tumor complexity. 

During recent years, artificial intelligence (AI) and machine learning (ML) technologies have been rapidly developed and applied for cancer diagnostic purposes. AI/ML technologies were mostly used in digital pathology in BC diagnosis. Some of the most effective AI-based techniques/programs including convolutional neural network (CNN), support vector machine (SVM), and random forest ML algorithms have been used to classify and distinguish normal and abnormal BC cells. ML-based programs have already been applied in some clinics to assist pathologists to improve the accuracy and efficiency for cancer diagnosis [22,23,24]. AI/ML models have also been applied to analyze histological and morphological features of ultrasound, histography, mammography, positron emission tomography (PET) and computerized tomography (CT) images, to identify malignant lesions and metastasis in lymph nodes [25]. To enhance the accuracy in cancer detection, different AI techniques are experimented with to obtain accurate classification of disease stages. Recently, AI/ML models have been developed and applied by integrating digital pathological images with the genomic and transcriptomic data in BC diagnosis and prognosis [24]. 

AI/ML may provide an effective approach with better precision than the pathological and histological prediction tools. Thus, AI/ML-based models may overcome the limitation of the existing diagnostic tools to improve their accuracy. However, AI-model based on single platform profiling such as imaging often fails to capture the complexity of the breast tumor. Currently, no gene mutation profile-based AI/ML model is developed to stratify and predict treatment response in BC. We have previously developed biomarker panel-based ML algorithms using gene expression profiles detected in primary tumor specimens and urine samples of large cohorts of prostate cancer patients and primary tumor of large colorectal cancer patients [23,26,27]. In this study, we aimed to apply our established ML methods to develop a gene mutations-based algorithm for predicting treatment response of BC patients who suffer different subtypes of cancer. 

## 2. Materials and Methods

### 2.1. Breast Cancer TCGA Cohort

For the breast cancer cohort of The Cancer Genome Atlas (TCGA) Firehose Legacy, data of 1108 invasive breast cancer patient tissue specimens were obtained from cBioportal. Among them, 399 specimens were obtained from treatment-naïve, invasive breast cancer (BC) patients collected at diagnosis [20,28,29,30] and was termed TCGA Cohort in this study. The prevalence and co-occurrence of genes previously shown to be altered in primary BC were similar in this more aggressive and advanced cohort as in other known BC cohorts. Alterations in *CDH1*, *TP53*, *GATA3*, *CCND1*, *PIK3CA*, and *PTEN* arose in patterns consistent with their known associations with specific receptor types and histology. All genomic and clinical data including gene alterations, gene mutations, patient age at diagnosis, cancer stage, tumor laterality, receptor status, cancer progression after the first-line chemotherapy, and overall survival during follow-up was obtained from cBioportal (https://www.cbioportal.org/study/summary?id=brca_tcga, accessed on 1 January 2022). In the 399 patient TCGA Cohort, 46 patients experienced cancer progression/recurrence after treatment, while 353 patients remained disease-free without progression/recurrence during follow-up. The receptor status of estrogen, progesterone and HER2 of each patient was collected. Forty-two patients had triple-negative BC. Among the triple-negative cancer patients, 7 developed cancer progression/recurrence after treatment while 35 remained disease-free. A total of 155 had luminal A BC, 13 experienced cancer progression/recurrence, while 142 remained disease-free after treatment. A total of 14 had luminal B, and 14 had HER2^+^ BC. 174 patients could not be classified in any BC subtype due to their HER2 status data being either missing or labeled “Equivocal” (Table 1).

### 2.2. Breast Cancer MSK Cohort

Breast cancer MSK Cohort from the Memorial Sloan Kettering was obtained from cBioportal (https://www.cbioportal.org/, accessed on 1 January 2022). Primary tumor specimens from treatment-naive BC patients (*n* = 1918) were collected at diagnosis and used for this study. Gene alterations, mutations, genomic profiling, and clinical data including patient age at diagnosis, cancer stage, tumor laterality, receptor status, cancer progression after first-line chemotherapy, and progression-free and overall survival during follow-up were extracted from cBioportal (https://www.cbioportal.org/study/summary?id=breast_msk_2018, accessed on 1 January 2022). These patients formed an MSK Cohort (*n* = 807), with 314 patients that developed cancer progression/recurrence after treatment and 493 patients being progression-free during follow-up. The receptor status of estrogen, progesterone and HER2 of each patient was obtained. In a total of 75 triple-negative patients, 29 developed cancer progression/recurrence, while 46 remained disease-free. In the 501 patients with luminal A cancer, 179 experienced cancer progression/recurrence, while 322 had no progression/recurrence (Table 1). In addition, 6 patients had luminal B, and 15 had HER2+ BC. Two hundred and ten patients could not be classified in any BC subtype since their HER2 status data was either missing or labeled “Equivocal” (Table 1). 

### 2.3. Machine Learning Algorithms

A random forest machine learning algorithm screening was performed to select combinations of mutation profiles of the genes in the RAS-RAF-MEK-ERK and PI3K/Akt/PTEN/mTOR pathways as well as genes frequently mutated in breast cancer, to form classifiers by using the established methods previously described [23,26]. Using the TCGA Cohort as a training set, the random forest algorithm classifiers, which combine different gene mutation profiles, were used to distinguish treatment responders (non-progression) and non-responders (progression) using XLSTAT software (Addinsoft, Paris, France). During the development of each random forest algorithm, the size of the forest was determined based on the number of patients in the cohort (>½ of the patient number). Each tree was developed from a bootstrap sample selected from the training data consisting of an arbitrary subset of genes. Confusion matrix of each random forest algorithm was used to identify the gene algorithm with the highest classification accuracy to distinguish responders and non-responders to first-line chemotherapy. The random forest parameters such as the number of trees were further tuned for the algorithm to optimize the accuracy and formed the final algorithm for classification of BC progression and response to treatment. In addition, gene mutation combinations with high accuracy were selected to perform 10-fold cross-validation in grid search to verify the classification performance and find the best gene combination. Among the algorithms of the gene combinations tested, a 12-Gene Algorithm consisting of mutation profiles of *KRAS*, *PIK3CA*, *MAP3K1*, *MAP2K4*, *PTEN*, *TP53*, *CDH1*, *GATA3*, *KMT2C*, *ARID1A*, *RunX1*, and *ESR1* showed the highest accuracy to distinguish responder and non-responder BC patients and was therefore chosen for this study. The 12-Gene Algorithm was further validated in the MSK Cohort using the same algorithm and cutoff value to classify responder and non-responder patients.

### 2.4. Statistical Analysis

To assess the accuracy of the 12-Gene Algorithm to distinguish progression and non-progression after treatment, logistic regression analysis was performed by comparing progression or non-progression classification by the algorithm with progression or non-progression data collected during follow-up for each sample. receiver operating characteristic (ROC) curve was plotted and area under the ROC curve (AUC) with 95% confidence interval (CI) was calculated using XLSTAT. Sensitivity, specificity, positive predictive value, negative predictive value and their respective 95% CI were calculated. In addition, discriminant analysis was conducted to test the predictive accuracy and the result was compared with that from logistic regression, as described previously [23,26]. The nonparametric Mann–Whitney test was performed to compare different groups. To measure the predictive power for cancer progression/recurrence after treatment, univariate and multivariate Cox proportional hazards regression analyses and Kaplan–Meier survival plot were conducted using XLSTAT. 

## 3. Results

### 3.1. Development of the 12-Gene Algorithm for Stratification of Responder and Non-Responder Patients to Predict Treatment Response

Disease progression after treatment is a major indicator of treatment response. Based on the clinical data of the TCGA Cohort, we divided the patients into two subgroups: (i) the responder group: patients had no disease progression after first-line chemotherapy during the 150 months follow-up period; (ii) the non-responder group: patients experienced disease progression after first-line chemotherapy during follow-up. We utilized a random forest ML classification screening to test if various combinations of mutation profiles of the candidate genes might be able to distinguish responders from non-responders in the total patient population or in patient subgroups, especially in those suffering triple-negative cancer, as currently no targeted drugs are available for effective treatment. Among all the gene mutation-based algorithms tested, an algorithm termed 12-Gene Algorithm using mutation profiles of *KRAS*, *PIK3CA*, *MAP3K1*, *MAP2K4*, *PTEN*, *TP53*, *CDH1*, *GATA3*, *KMT2C*, *ARID1A*, *RunX1*, and *ESR1* exhibited the highest accuracy and was chosen as the classifier (Figure 1). 

As determined using logistic regression analysis, the 12-Gene Algorithm had an accuracy at distinguishing responders from non-responders in the TCGA Cohort with AUC of 0.96 (95% CI 0.94–0.98), sensitivity of 72% (95% CI 59–85%) and specificity of 97% (95% CI 95–98%) (*p* < 0.0001; Table 2, Appendix A, Figure 2A). We examined the performance of cancer stage, a well-established clinical and pathological risk indicator in the same cohort. Cancer stage was not able to identify non-responders, with a lower AUC of 0.66 (95% CI 0.59–0.74) and lower sensitivity of 11% (95% CI 1.9–20%) than that of the 12-Gene Algorithm (Table 2, Figure 2B), suggesting that cancer stage cannot be used as a biomarker to stratify treatment response of BC patients. Moreover, when the 12-Gene Algorithm was combined with cancer stage, the accuracy was similar as compared with the 12-Gene Algorithm with AUC of 0.93 (95% CI 0.90–0.96), sensitivity of 78% (95% CI 66–90%), and specificity of 96% (95% CI 94–98%) (Table 2, Figure 2C). These data suggest that the 12-Gene Algorithm alone may be used as a classifier to distinguish responder from non-responder BC patients after the first-line chemotherapy.

Currently, there is no targeted drugs and predictive biomarkers for treatment response for patients suffering tripe-negative BC, we therefore examined the performance of the 12-Gene Algorithm as a classifier to distinguish responders from non-responders suffering triple-negative BC. Interestingly, similar to what was observed in the cohort with all patients, the 12-Gene Algorithm had high accuracy in triple-negative cancer patients (*n* = 42) with AUC of 0.85 (95% CI 0.73–0.98), sensitivity of 71% (95% CI 38–105%) and specificity of 97% (95% CI 92–103%) (Table 2 and Appendix A, Figure 2D). In contrast, cancer stage showed AUC of 0.54 (95% CI 0.30–0.77) with 0% sensitivity, suggesting that cancer stage cannot be used as a classifier to differentiate responders and non-responders in triple-negative cancer (Table 2, Figure 2E). When the 12-Gene Algorithm was combined with cancer stage, the sensitivity and AUC values increased with AUC of 0.98 (95% CI 0.93–1.02) and sensitivity of 91% (95% CI 82–101%) (Table 2, Figure 2F). Taken together, our data suggest that the 12-Gene Algorithm may be used as an independent classifier to distinguish responder from non-responder triple-negative BC patients after the first-line chemotherapy. 

To further test the accuracy of the 12-Gene Algorithm in other subtype of BC, we assessed the algorithm in luminal A-subtype of BC within the TCGA Cohort. Logistic regression analysis revealed the performance of the 12-Gene Algorithm with AUC of 0.89 (95% CI 0.83–0.96), sensitivity of 85% (95% CI 65–104%) and specificity of 96% (95% CI 93–100%), showing high accuracy at distinguishing responders from non-responders in liminal A BC (*n* = 155) (Table 2, Appendix A, Figure 2G). In contrast, cancer stage cannot be used as a classifier in luminal A BC, as shown by AUC of 0.64 (95% CI 0.49–0.78) and sensitivity of 0% (Table 2, Figure 2H). When they were combined, the performance was similar with AUC of 0.94 (95% CI 0.89–0.98) (Table 2, Figure 2I). Again, the data showed that the 12-Gene Algorithm may serve as an independent classifier to distinguish responder from non-responder luminal A patients.

### 3.2. Assessment of the 12-Gene Algorithm for Prediction of Progression-Free Survival after First-Line Therapy in the TCGA Cohort

To assessed whether the 12-Gene Algorithm might be used as a biomarker to predict progression-free survival (PFS) in the TCGA Cohort, Kaplan–Meier plot and log-rank analysis were performed. 

We observed that there was a large and statistically significant difference in PFS between the subgroups stratified by the 12-Gene Algorithm classification scores. Patients with high 12-Gene Algorithm scores in their primary tumors had significantly poorer PFS compared with those with low scores (log-rank *p* < 0.0001, Figure 3A). There was a small but statistically significant difference in PFS between the subgroups stratified by cancer stage (Stage I/II vs. III/IV) (*p* = 0.000, Figure 3B). 

In clinical practice, there is no predictive biomarker available to predict treatment response for triple-negative patients, we therefore wanted to examine whether the 12-Gene Algorithm might be used to predict treatment response in triple-negative BC in the TCGA Cohort. Indeed, triple-negative BC patients who had higher scores of the 12-Gene Algorithm in their tumors had significantly poorer PFS as compared with those who had lower scores (*p* < 0.0001, Figure 3C). However, there was no significant differences in PFS between triple-negative patients with higher and lower cancer stage (Stage I/II vs. III/IV) (*p* = 0.224, Figure 3D). We further assessed the 12-Gene Algorithm in luminal A patients. Indeed, the luminal-A patients who had higher 12-Gene Algorithm scores experienced significantly poorer PFS as compared with lower score patients (*p* < 0.0001, Figure 3E). However, there was no significant difference in PFS between two cancer stage groups (*p* = 0.068, Figure 3F).

### 3.3. The 12-Gene Algorithm as a Predictive Biomarker for Treatment Response in the TCGA Cohort

To further assess the ability of the 12-Gene Algorithm for predicting treatment response, we performed univariate and multivariate Cox proportional hazard regression analyses. The univariate analysis revealed high predictive power of the 12-Gene Algorithm as indicated by hazard ratio (HR) of 21.6 (95% CI 11.3–41.5, *p* < 0.0001; Table 3), while the HR value for cancer stage was only 2.8 (95% CI 1.6–5.1, *p* < 0.001; Table 3). The multivariate Cox analysis showed that HR of the 12-Gene Algorithm reached 19.7 (95% CI 10.2–38.1, *p* < 0.0001; Table 3), while HR for cancer stage was 1.9 (95% CI 1.1–3.5, *p* = 0.031; Table 3). These data showed that the 12-Gene Algorithm had higher predictive power for PFS than clinicopathological factor such as cancer stage. 

To assess the predictive power of the 12-Gene Algorithm in triple-negative BC patients, we performed univariate and multivariate Cox proportional hazard regression analyses. As in all BC patients, the 12-Gene Algorithm had a high HR of= 19.3 (95% CI 3.7–101.3, *p* = 0.000) in univariate and HR of 22.3 (95% CI 4.0–125.7, *p* = 0.000) in multivariate Cox analyses in triple-negative BC patients (Table 3). In contrast, cancer stage had lower HR of 2.7 (95% CI 0.52–13.8; *p* = 0.242) in univariate and 3.8 (95% CI 0.62–22.9; *p* = 0.151) in multivariate Cox analyses (Table 3). The result showed the 12-Gene Algorithm as an independent biomarker to accurately predict treatment response for triple-negative BC. 

A total of 155 patients in the TCGA Cohort had luminal A subtype of BC, we wanted to examine whether the 12-Gene Algorithm may predict PFS in luminal A patients. Univariate Cox proportional hazard regression analyses revealed that the 12-Gene Algorithm had high predictive power with HR of 47.6 (95% CI 10.4–217.0, *p* < 0.0001) in univariate and HR of 45.4 (95% CI 9.6–214.5, *p* < 0.0001) in multivariate Cox analyses (Table 3). In contrast, cancer stage had much lower HR in both univariate and multivariate Cox analysis (Table 3). Taken together, the 12-Gene Algorithm was demonstrated to have high predictive power for treatment response in BC patients in the TCGA Cohort, and more importantly in triple-negative and luminal A subgroups of BC.

### 3.4. Validation of the 7-Gene Algorithm in the MSK Cohort

The fully trained model of the 12-Gene Algorithm was tested for validation in an independent external cohort of 807 patients as MSK Cohort (Figure 1 and Table 1). 

In this cohort, 314 out of 807 patients did not respond to treatment, whereas 493 patients responded to treatment with no progression/recurrence. The same random forest machine learning algorithm using mutation profiles of the 12 genes as developed in the TCGA Cohort was used to classify each patient in the MSK Cohort as a treatment responder without progression or treatment non-responder with progression. Logistic regression analysis was conducted, and the results showed that the 12-Gene Algorithm exhibited high accuracy at distinguishing responder and non-responder patient groups with an AUC of 0.97 (95% CI 0.96–0.98), sensitivity of 75% (95% CI 70–79%) and specificity of 97% (95% CI 96–99%) (Table 4 and Appendix A, Figure 4A). Similar to what was observed in the TCGA Cohort, cancer stage could not identify non-responders with a sensitivity of 0% (95% CI 0–0%), and AUC of 0.83 (95% CI 0.80–0.86) (Table 4, Figure 4B). Combining the 12-Gene Algorithm with cancer stage did not improve its accuracy (Table 4, Figure 4C). 

More importantly, the 12-Gene Algorithm exhibited high accuracy at distinguishing responder from non-responder patients who suffered triple-negative BC (*n* = 75) in the MSK Cohort with AUC of 0.981 (95% CI 0.95–1.01), sensitivity of 90% (95% CI 79–101%) and specificity of 91% (95% CI 83–99%) (Table 4 and Appendix A, Figure 4D). Such accuracy was higher than what was observed in the TCGA training cohort. In contrast, cancer stage showed 0% sensitivity, suggesting that cancer stage cannot be used to distinguish responders and non-responders in triple-negative BC subgroup (Table 4, Figure 4E). In the MSK cohort, 501 patients had Luminal A-subtype of BC. Similar to what was shown in the TCGA Cohort, the 12-Gene Algorithm exhibited high accuracy at distinguishing responder and non-responder luminal A patients with AUC of 0.98 (95% CI 0.96–0.99), sensitivity of 73% (95% CI 67–80%), and specificity of 99% (95% CI 97–100%) (Table 4 and Appendix A, Figure 4G). However, cancer stage had AUC of 0.84 (95% CI 0.81–0.88) and 0% sensitivity (Table 4, Figure 4H), showing no ability to stratify responders and non-responders in luminal A BC. 

### 3.5. Assessment of the 12-Gene Algorithm as a Predictive Biomarker for PFS in the MSK Cohort

Kaplan–Meier and log-rank analysis were performed to assess the predictive power of the 12-Gene Algorithm in the MSK Cohort. There was a statistically significant difference in PFS between the subgroups stratified by the 12-Gene Algorithm scores in all patients in the MSK Cohort (*p* < 0.0001, Figure 5A). However, there was no statistically significant differences in PFS between high and low cancer stage patient groups (*p* = 0.069; Figure 5B). Further, in triple-negative BC in the MSK Cohort, patient groups with high and low 12-Gene Algorithm scores had a large and statistically significant difference in PFS (*p* < 0.0001, Figure 5C), while no statistically significant difference in PFS was found between two cancer stage groups (*p* = 0.814, Figure 5D). In the Luminal A subgroup, patient groups with higher and lower 12-Gene Algorithm scores exhibited larger and statistically more significant difference in PFS (*p* < 0.0001, Figure 5E) as compared with PFS difference between cancer stage I/II and III/IV groups (*p* = 0.033, Figure 5F).

### 3.6. Validation of the 12-Gene Algorithm as an Independent PFS Predictor in the MSK Cohort

We further validated the predictive power of the 12-Gene Algorithm in the MSK Cohort. The univariate Cox regression analysis showed higher predictive power of the 12-Gene Algorithm with HR of 4.4 (95% CI 3.4–5.7; *p* < 0.0001) than cancer stage with HR of 1.3 (95% CI 0.8–1.7; *p* = 0.072) (Table 5). A similar result was observed in the multivariate Cox analyses. 

Next, we validated the predictive power of the 12-Gene Algorithm in triple-negative BC patients in the MSK Cohort (*n* = 75). Cox regression showed high predictive power with HR of 18.6 (95% CI 4.4–79.2; *p* < 0.0001) in univariate and 22.4 (95% CI 4.9–103.2; *p* < 0.0001) in multivariate analysis, which was similar to the results in the TCGA Cohort (Table 5). Cancer stage had much lower predictive power with low HR in both univariate and multivariate Cox analysis in triple-negative patients (Table 5). The data suggest the potential utilization of the 12-Gene Algorithm as a predictive biomarker for cancer progression after treatment in patients with triple-negative BC.

Univariate and multivariate Cox regression analyses were performed to assess the ability of the 12-Gene Algorithm to predict treatment response in luminal A subgroup. The 12-Gene Algorithm had a HR of 3.8 (95% CI 2.7–5.4, *p* < 0.0001) in univariate analysis and 3.7 (95% CI 2.6–5.4, *p* < 0.0001) in multivariate analysis in the MSK Cohort (Table 5). In contrast, cancer stage had lower HR in Luminal A patients (Table 5).

## 4. Discussion

Previous reported studies have shown that ML models based on the imaging data can be used for improving clinical diagnosis. However, only limited studies using ML technologies to develop prognostic biomarkers have been reported. In this study, we developed and validated a ML technology-based 12-Gene Algorithm by using a large amount of cancer genomics/gene mutation data from two large patient cohorts. We showed that the ML algorithm of the 12-Gene mutation profiles had high accuracy of performance for stratification and prediction of treatment response in patients who suffered triple-negative cancer or luminal A cancer. We showed that the 12-Gene Algorithm could be used to distinguish BC patients with high risk of not responding to first-line chemotherapies. The novelty of this finding includes: (i) applying artificial Intelligence/ML modeling for biomarker identification by screening well-known mutations of genes that are involved in BC progression; (ii) building a predictive a model/algorithm based on multiple gene mutation profiles, which serves as a robust biomarker to predict treatment response of patients with BC, especially for patients with triple-negative BC, which lacks a reliable biomarker for the prediction of the treatment response; (iii) showing the 12-Gene Algorithm with consistently high predictive accuracy in two independent cohorts, which demonstrated its potential as a robust and reliable gene mutation-based test. 

Triple-negative or basal-like BC subtype is associated with shorter time to recurrence and higher metastatic potential [2,3,4], mostly due to that the breast tumors have features of heterogeneity and complexity, and harbor multiple gene mutations. Although ER, PR and HER2 are commonly used for BC subtyping, especially for diagnosis and treatment decisions, they are not robust for prediction of treatment response, especially for triple-negative BC patients. In this study, one of our novel findings is that the 12-Gene Algorithm was used as an accurate biomarker to distinguish triple-negative BC patients who were non-responders and responders to first-line chemotherapies in both TCGA Cohort and MSK Cohort. Moreover, the 12-Gene Algorithm showed similarly high accurate performance to distinguish treatment response in luminal A patients in both cohorts. More importantly, we used univariate and multivariate Cox regression analyses to test the predictive power of the 12-Gene Algorithm in triple-negative BC patients from TCGA and MSK cohorts. Our results showed that the 12-Gene Algorithm had high HR with statistical significance to predict PFS in triple-negative BC patients and the results are highly consistent in the two independent cohorts. In addition, the 12-Gene Algorithm had high HR with statistical significance in luminal A patients in the TCGA Cohort. Interestingly, the HR of the 12-Gene Algorithm to predict PFS of luminal A patients in the MSK Cohort was lower as compared to that in the TCGA Cohort. This may be attributed to that luminal A subtype may have more heterogenous features in these two cohorts. 

Since currently no targeted therapy is available for effective treatment of triple-negative BC, the 12-Gene Algorithm that is based on multiple gene mutation status which reflects on the complexity of the cancer genome of this subtype, may offer a novel tool for tailored treatment decision and prediction of treatment outcome. Our results suggest that there is a potential to further develop the 12-Gene Algorithm for validation in clinical settings to aid the patient stratification and treatment decision-making for newly diagnosed BC patients.

In this study, we found high prevalence of mutations of *KRAS*, *PIK3CA*, *MAP3K1*, *MAP2K4*, *PTEN*, *TP53*, *CDH1*, *GATA3*, *KMT2C*, *ARID1A*, *RunX1*, and *ESR1* genes occurring in primary tumor specimens of BC patients, including triple-negative and luminal A patients. The molecular mechanisms and functional studies in cell line- and animal-based models have shown that most of the 12 genes in the algorithm play key roles in cancer metastasis and treatment resistance in BC. However, none of the clinical risk scores or gene mutation tests using some of the 12 genes, such as *KRAS* and its related genes, had achieved high prognostic accuracy and reliability to justify clinical application [26]. To date, no gene mutation test is able to predict PFS in BC cohorts with high power. Our finding suggests that the ML-based 12-Gene Algorithm by combining gene mutation profiles of 12 genes, achieved high accuracy in predicting BC treatment response. Thus, our study proposed a new biomarker tool for improvement of BC diagnosis and prognosis. 

Developing ML-based algorithms for disease diagnosis and prognosis has emerged as a useful tool. The Multi-omic ML model, a simplified reproducible predictive model, showed the highest predictive power as compared with other models [22]. However, this ML-model is based on data from a defined small patient cohort and has not been validated in different patient cohorts. Our 12-Gene Algorithm in this study has advantages due to that the cancer genomic data are not affected by pathologically observations, and this algorithm was identified through high-throughput screening of large numbers of algorithms based on combinations of different gene mutation profiles. Finally, the algorithm has shown predictive performance with high accuracy in two independent triple-negative BC patient populations and two independent luminal A patient populations. Our finding suggests that using ML to develop diagnostic or prognostic algorithms is a viable approach, which may provide a direction and advanced tool for improving future development of cancer diagnostics and prognostics. 

Although the 12-Gene algorithm showed high performance at predicting BC treatment response, there are some limitations in this study. First, we did not have prospective cohorts to validate the algorithm. Second, patients from more conventional clinical settings need to be included in the study to avoid potential bias. In the future, more studies, especially prospective studies at multiple clinical settings, will be conducted to further validate the 12-Gene Algorithm in large patient populations including triple-negative BC patients. 

## 5. Conclusions

In conclusion, we developed and validated a novel 12-Gene Algorithm for predicting BC treatment response in newly diagnosed patients. It showed higher accuracy than clinicopathological factors such as cancer stage in side-by-side assessment in two independent BC cohorts. It has a potential to be used in clinical practice to improve BC treatment decision-making, especially for triple-negative BC patients.

## Figures and Tables

**Figure 1 cancers-14-05322-f001:**
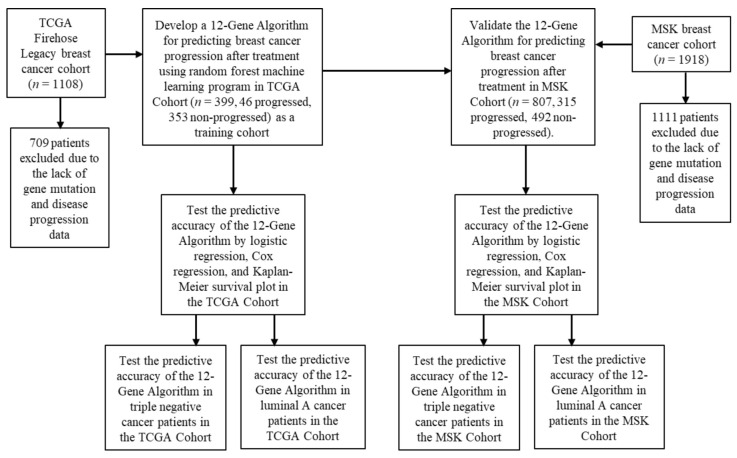
Study design.

**Figure 2 cancers-14-05322-f002:**
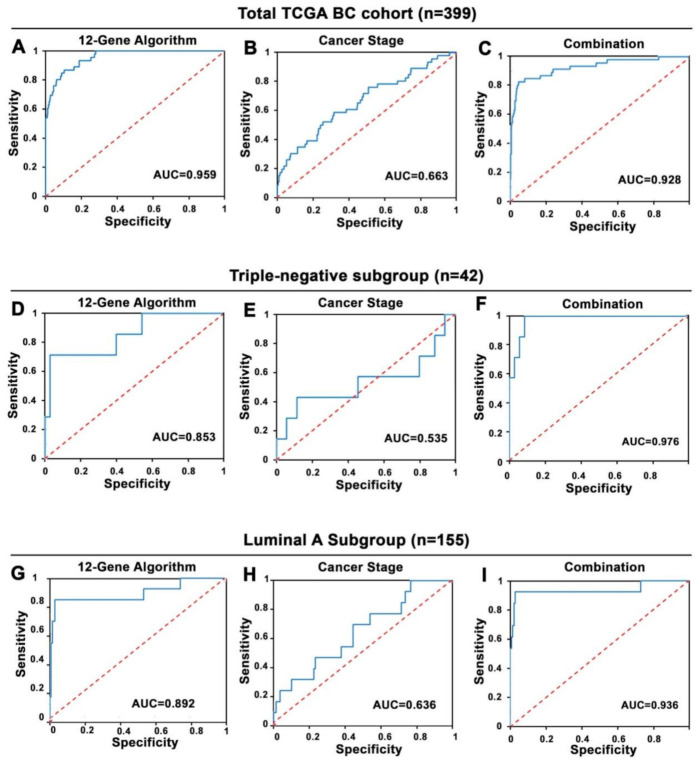
Receiver operating characteristic (ROC) curves of the 12-Gene Algorithm and cancer stage for distinguishing treatment responder and non-responder patients in the TCGA Cohort. (**A**–**C**) ROC curves of the 12-Gene Algorithm in (**A**), cancer stage in (**B**), and their combination in (**C**) in the TCGA Cohort (*n* = 399). (**D**–**F**) ROC curves of the 12-Gene Algorithm in (**D**), cancer stage in (**E**), and their combination in (**F**) in the triple-negative patients (*n* = 42) in the TCGA Cohort. (**G**–**I**) ROC curves of the 12-Gene Algorithm in (**G**), cancer stage in (**H**), and their combination in (**I**) in luminal A patients (*n* = 155) in the TCGA Cohort. Areas under the ROC curve (AUC) values are shown.

**Figure 3 cancers-14-05322-f003:**
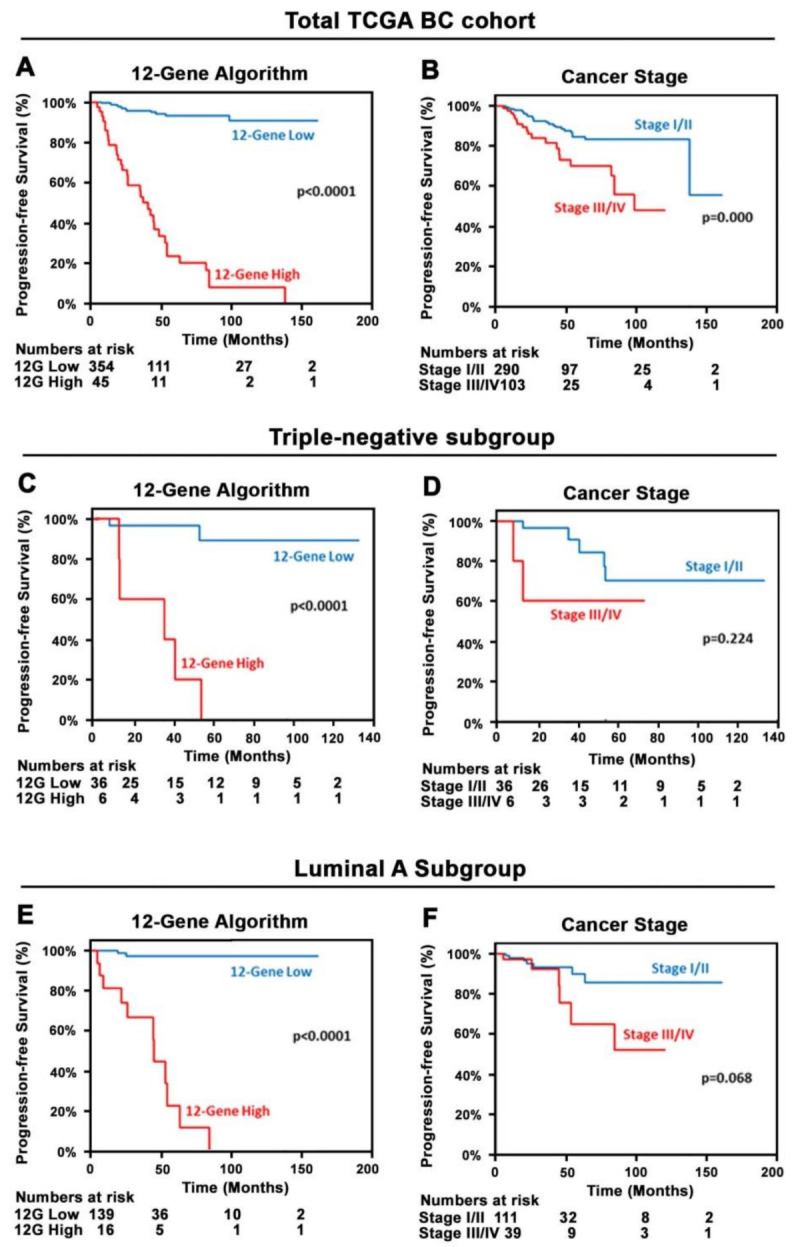
Kaplan–Meier survival analyses of the 12-Gene Algorithm and cancer stage for predicting PFS in the TCGA Cohort. (**A**,**B**) Kaplan–Meier survival curves of the 12-Gene Algorithm in (**A**) and cancer stage in (**B**) in the TCGA Cohort (*n* = 399). (**C**,**D**) Kaplan–Meier survival curves of the 12-Gene Algorithm in (**C**), cancer stage in (**E**) in triple-negative patients (*n* = 42) in the TCGA Cohort. (**E**,**F**) Kaplan–Meier survival curves of the 12-Gene Algorithm in (**E**) and cancer stage in (**F**) in luminal A patients (*n* = 155) in the TCGA Cohort. Log rank *p* values are shown.

**Figure 4 cancers-14-05322-f004:**
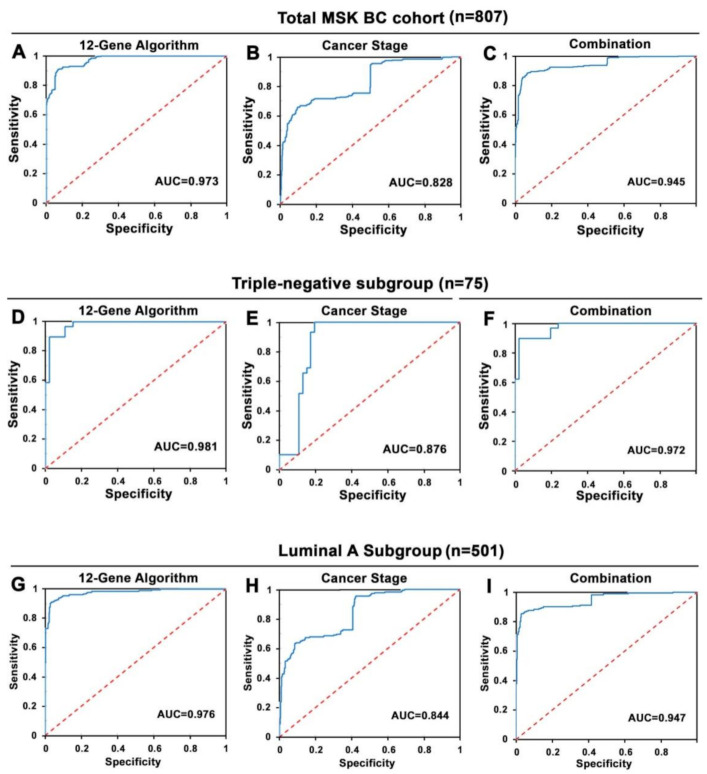
Receiver operating characteristic (ROC) curves of the 12-Gene Algorithm and cancer stage for distinguishing responder and non-responder patients in the MSK Cohort. (**A**–**C**) ROC curves of the 12-Gene Algorithm in (**A**), cancer stage in (**B**), and their combination in (**C**) in all patients in the MSK Cohort (*n* = 807). (**D**–**F**) ROC curves of the 12-Gene Algorithm in (**D**), cancer stage in (**E**), and their combination in (**F**) in triple-negative patients (*n* = 75) in the MSK Cohort. (**G**–**I**) ROC curves of the 12-Gene Algorithm in (**G**), cancer stage in (**H**), and their combination in (**I**) in luminal A patients (*n* = 501) in the MSK Cohort. Areas under the ROC curve (AUC) values are shown.

**Figure 5 cancers-14-05322-f005:**
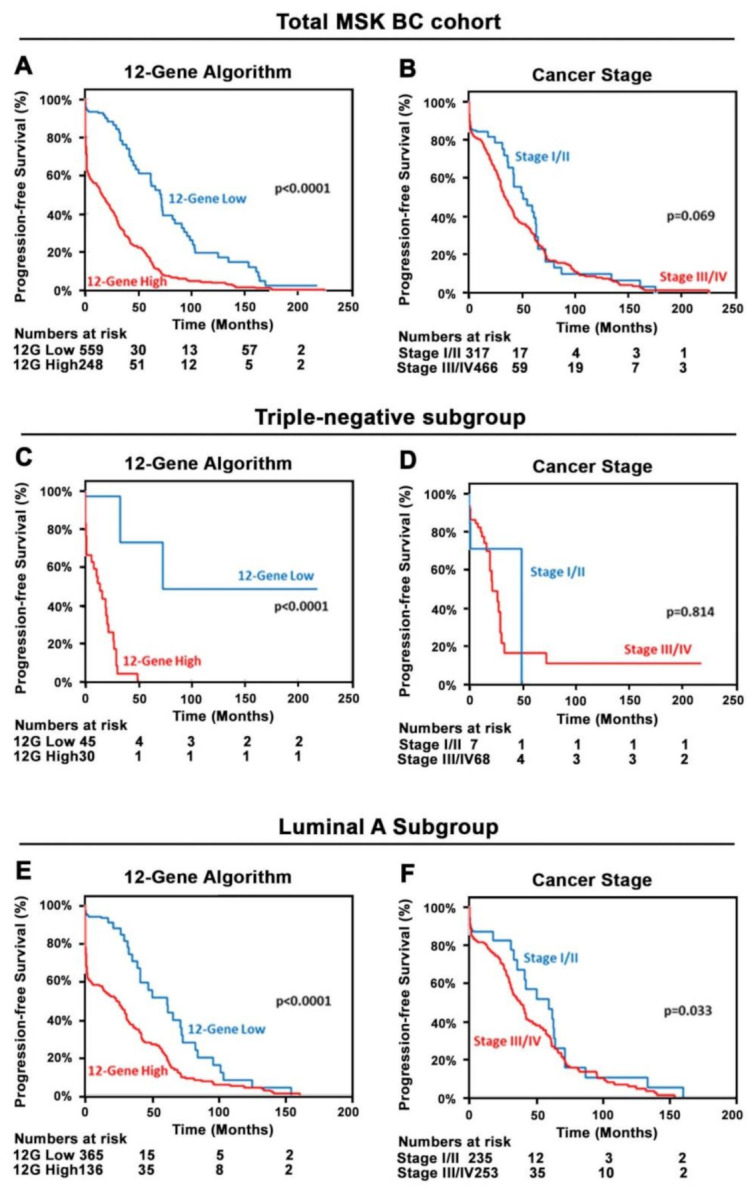
Kaplan–Meier survival analyses of the 12-Gene Algorithm and cancer stage for predicting progression-free survival in the MSK Cohort (*n* = 807). (**A**,**B**) Kaplan–Meier survival curves of the 12-Gene Algorithm in (**A**) and cancer stage in (**B**) in the MSK Cohort (*n* = 807). (**C**,**D**) Kaplan–Meier survival curves of the 12-Gene Algorithm in (**C**) and cancer stage in (**D**) in triple-negative patients (*n* = 75) in the MSK Cohort. (**E**,**F**) Kaplan–Meier survival curves of the 12-Gene Algorithm in (**E**) and cancer stage in (**F**) in luminal A patients (*n* = 501) in the MSK Cohort. Log rank *p* values are shown.

**Table 1 cancers-14-05322-t001:** Clinical characteristics of BC patients from TCGA and MSK cohorts.

	TCGA Cohort	MSK Cohort
No of patients	399	807
Median age at diagnosis (Q1, Q3)	59 (49, 68)	54 (46, 65)
Cancers stage at diagnosis (%)		
Stage I	50 (13%)	342 (42%)
Stage II	240 (60%)	232 (29%)
Stage III	98 (25%)	99 (12%)
Stage IV	5 (1%)	134 (17%)
Unknown	6 (2%)	0
Tumor laterality (%)		
Left side	213 (53%)	420 (52%)
Right side	186 (47%)	387 (48%)
Triple-negative cancer (%)	42 (11%)	75 (9%)
Luminal A cancer (%)	155 (39%)	501 (62%)
Luminal B cancer (%)	14 (4%)	6 (0.7%)
HER2+ cancer (%)	14 (4%)	15 (1.9%)
Cancer type unknown	174 (44%)	210 (26%)
Overall survival (%)		
Living	377 (94%)	713 (88%)
Diseased	22 (6%)	94 (12%)
Progression/recurrence after treatment (%)		
Progressed/recurrent	46 (12%)	314 (39%)
Non-progressed/non-recurrent	353 (88%)	493 (61%)

**Table 2 cancers-14-05322-t002:** Accuracy of the 12-Gene Algorithm and cancer stage for distinguishing treatment responders and non-responders in the TCGA Cohort (*n* = 399), triple-negative subgroup (*n* = 42), and luminal A subgroup (*n* = 155).

	Sensitivity (95% CI)	Specificity (95% CI)	PPV (95% CI)	NPV (95% CI)
**TCGA Cohort (*n* = 399)**				
12-Gene Algorithm	72% (59–85%)	97% (95–98%)	73% (60–86%)	97% (95–98%)
Cancer stage	11% (1.9–20%)	99% (99–100%)	71% (38–105%)	89% (86–92%)
Combination	78% (66–90%)	96% (94–98%)	73% (61–86%)	97% (95–99%)
**Triple-negative breast cancer in the TCGA Cohort (*n* = 42)**				
12-Gene Algorithm	71% (38–105%)	97% (92–103%)	83% (54–113%)	94% (87–102%)
Cancer stage	0% (0–0%)	100% (100–100%)	0% (0–0%)	83% (72–95%)
Combination	91% (82–101%)	100% (100–100%)	70% (42–98%)	100% (100–100%)
**Luminal A breast cancer in the TCGA Cohort (*n* = 155)**				
12-Gene Algorithm	85% (65–104%)	96% (93–100%)	69% (46–91%)	99% (97–101%)
Cancer stage	0% (0–0%)	100% (100–100%)	0% (0–0%)	91% (87–96%)
Combination	85% (65–104%)	97% (94–100%)	73% (51–96%)	99% (97–101%)

CI: confidence interval; PPV: positive predictive value; NPV: negative predictive value.

**Table 3 cancers-14-05322-t003:** Univariate and multivariate Cox regression analyses of the 12-Gene Algorithm and cancer stage for predicting cancer progression after treatment in the TCGA Cohort (*n* = 399), triple-negative subgroup (*n* = 42), and luminal A subgroup (*n* = 155).

	Univariate		Multivariate	
	HR (95% CI)	*p* Value	HR (95% CI)	*p* Value
**TCGA Cohort (*n* = 399)**				
12-Gene Algorithm	21.6 (11.3–41.5)	<0.0001	19.7 (10.2–38.1)	<0.0001
Cancer stage	2.8 (1.6–5.1)	<0.001	1.9 (1.1–3.5)	0.031
**Triple-negative breast cancer in the TCGA Cohort (*n* = 42)**				
12-Gene Algorithm	19.3 (3.7–101.3)	0.000	22.3 (4.0–125.7)	0.000
Cancer stage	2.7 (0.52–13.8)	0.242	3.8 (0.62–22.9)	0.151
**Luminal A breast cancer in the TCGA Cohort (*n* = 155)**				
12-Gene Algorithm	47.6 (10.4–217.0)	<0.0001	45.4 (9.6–214.5)	<0.0001
Cancer stage	2.7 (0.89–7.9)	0.080	1.1 (0.36–3.6)	0.835

**Table 4 cancers-14-05322-t004:** Accuracy of the 12-Gene Algorithm and cancer stage for distinguishing progression and non-progression after treatment in the MSK Cohort (*n* = 807), triple-negative subgroup (*n* = 75), and luminal A subgroups (*n* = 501).

	Sensitivity (95% CI)	Specificity (95% CI)	PPV (95% CI)	NPV (95% CI)
**MSK Cohort (*n* = 807)**				
12-Gene Algorithm	75% (70–79%)	97% (96–99%)	95% (92–98%)	86% (83–89%)
Cancer stage	0% (0–0%)	100% (100–100%)	0% (0–0%)	62% (58–65%)
Combination	75% (70–80%)	97% (96–99%)	95% (92–97%)	86% (83–89%)
**Triple-negative breast cancer in the MSK Cohort (*n* = 75)**				
12-Gene Algorithm	90% (79–101%)	91% (83–99%)	87% (75–99%)	93% (86–101%)
Cancer stage	0% (0–0%)	100% (100–100%)	0% (0–0%)	61% (50–72%)
Combination	90% (79–101%)	91% (83–99%)	87% (75–99%)	93% (86–101%)
**Luminal A breast cancer in the MSK Cohort (*n* = 501)**				
12-Gene Algorithm	73% (67–80%)	99% (97–100%)	96% (93–99%)	87% (83–90%)
Cancer stage	0% (0–0%)	100% (100–100%)	0% (0–0%)	65% (61–69%)
Combination	73% (66–80%)	98% (97–100%)	96% (93–99%)	87% (84–91%)

CI: confidence interval; PPV: positive predictive value; NPV: negative predictive value.

**Table 5 cancers-14-05322-t005:** Univariate and multivariate Cox regression analyses of the 12-Gene Algorithm and cancer stage for predicting treatment response in the MSK Cohort (*n* = 807), triple-negative subgroup (*n* = 75), and luminal A subgroup (*n* = 501).

	Univariate		Multivariate	
	HR (95% CI)	*p* Value	HR (95% CI)	*p* Value
**MSK Cohort (*n* = 807)**				
12-Gene Algorithm	4.4 (3.4–5.7)	<0.0001	4.4 (3.3–5.7)	<0.0001
Cancer stage	1.3 (0.8–1.7)	0.072	1.2 (1.0–1.6)	0.100
**Triple-negative breast cancer in the MSK Cohort (*n* = 75)**				
12-Gene Algorithm	18.6 (4.4–79.2)	<0.0001	22.4 (4.9–103.2)	<0.0001
Cancer stage	0.87 (0.26–2.9)	0.815	2.3 (0.56–10.7)	0.285
**Luminal A breast cancer in the MSK Cohort (*n* = 501)**				
12-Gene Algorithm	3.8 (2.7–5.4)	<0.0001	3.7 (2.6–5.4)	<0.0001
Cancer stage	1.5 (1.0–2.2)	0.035	1.3 (0.94–1.9)	0.104

HR: hazard ratio; CI: confidence interval.

## Data Availability

The data supporting reported results can be found in the publicly archived datasets analyzed or generated during the study.

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
