# Peer review of "K-RAS Associated Gene-Mutation-Based Algorithm for Prediction of Treatment Response of Patients with Subtypes of Breast Cancer and Especially Triple-Negative Cancer"

_cancers, 2022, doi:10.3390/cancers14215322_

Round 1

Reviewer 1 Report

The manuscript reports on a retrospective evaluation of a 12-gene mutation score to predict treatment response in early breast cancer (BC). Treatment response is intended as progression-free survival after (neo)adjuvant chemotherapy.

The findings are interesting, and the authors acknowledge limitations of their work, in particular the retrospective nature of the analysis. Thus, this predictive algorithm could be tested prospectively as it potentially addresses an unmet medical need

MAJOR OBSERVATIONS

The findings are retrospective in nature and based upon a small sample size, which becomes even smaller when the analysis is restricted to luminal A or TNBC patients. It is also unclear to this reviewer if the predictive value of the 12 gene mutation score is unaffected by the type of chemotherapy administered to individual patient (e.g., taxane vs platinum in TNBC). No information is available as to whether this patient population is homogeneous from this standpoint. It is this reviewer’s understanding that also early stage luminal A BC included in this analysis received chemotherapy.

Therefore, in abstract, page 2, line 47 this reviewer thinks that “great potential” could be an overstatement and suggests to remove “great”

In general, it would be valuable for the reader to have an example of how many BC patients (e.g., out of a total of 100) candidates to (neo)adjuvant chemotherapy will have a correct prediction of their response to treatment based upon the sensitivity and specificity of the test, or, in other words, how many false negatives and false positives there would be. This simulation would provide clinicians with a better appreciation of the potential of this algorithm in clinical practice  

MINOR OBSERVATIONS

Materials and Methods, Table 1 on page 4: TCGA cohort consists of 399 patients but when they are divided by BC type the sum is only 225 (42+155+14+14); the same for MSK cohort, instead of 807 patients the sum of different BC types is 597 (75+501+6+15)

Results page 6 line 237 please remove “Figure 1” on top of the figure, same for figure 2 (page 8), 3 (page 9), figure 4 on page 12 and figure 5 on page 14

Results page 7 please insert an empty row between rows 257 and 258

In many instances, results reported in tables and/or figures do not match with results in article body, or sometimes numbers appear to be rounded. This reviewer invites the authors to carefully double check each and every value reported in this article. Here below a non exhaustive list of such discrepancies:

Results page 7 Table 2 when numbers do not always match those in Figure 2 on page 8: line 278, AUC of 0.89 but in Fig. 2 panel G AUC is 0.959; line 285, 69% appears to be 97% in Table 2

Results figure 3 page 9: panel B “p=0.000” but on line 315 the manuscript reads “p<0.001”

Results table 3 HR for multivariate analysis in TNBC for cancer stage is 2.3 (0.36-14.0) with a p value of 0.382 but text on page 11 line 357 reads 3.8 (0.62-22.9) with a p value of 0.151

Results table 3 univariate analysis for 12-gene algorithm in luminal A has a p value of p<0.0001 but text on page 11, line 365 reads p<0.001; the same for multivariate analysis (table vs line 369 on page 11)

Results table 3 HR for multivariate analysis cancer stage is 1.2 but text on page 11 line 371 reads 1.1 and the p values are 0.763 in table 3 and 0.835 in text on page 11, line 371

Results page 11, lines 356 and 359 “tripe-negative” should be “triple-negative”

Results page 12 line 390 0.97 should be 0.973

Results table 4 panel E and text page 13 line 408 authors should specify if numbers are rounded (0.876 vs 0.88, respectively)

Results page 13 lines 405 and 412 and figure 4D: please uniform to either 0.981 or 0.982

References page 19 please remove 1. below ref nr. 30

Author Response

To: Cancers, 

RE: Manuscript ID, cancers-1912723

Dear Editors and Reviewers,

We sincerely thank the Editors and Reviewers for providing valuable comments and suggestions in helping us improve the quality of our manuscript. We are especially encouraged by the facts that both Reviewers found our findings “interesting” and “very meaningful”. In the revised manuscript, we have addressed the important points by specifically following the instructions provided by the Editors and Reviewers. In addition, we have performed English writing revisions to improve the language and writing style of the manuscript, as suggested by the Editors and Reviewers. Your invaluable comments and suggestions have guided us to improve the quality of our manuscript so it can meet the high standards set by the journal. The revised text is marked in red. Below are our specific, point-by-point responses addressing the important suggestions of the Reviewers.

Response to Reviewer 1:

Reviewer 1: Comments and Suggestions for Authors

The manuscript reports on a retrospective evaluation of a 12-gene mutation score to predict treatment response in early breast cancer (BC). Treatment response is intended as progression-free survival after (neo)adjuvant chemotherapy.

The findings are interesting, and the authors acknowledge limitations of their work, in particular the retrospective nature of the analysis. Thus, this predictive algorithm could be tested prospectively as it potentially addresses an unmet medical need.

MAJOR OBSERVATIONS

The findings are retrospective in nature and based upon a small sample size, which becomes even smaller when the analysis is restricted to luminal A or TNBC patients. It is also unclear to this reviewer if the predictive value of the 12 gene mutation score is unaffected by the type of chemotherapy administered to individual patient (e.g., taxane vs platinum in TNBC). No information is available as to whether this patient population is homogeneous from this standpoint. It is this reviewer’s understanding that also early stage luminal A BC included in this analysis received chemotherapy.

Therefore, in abstract, page 2, line 47 this reviewer thinks that “great potential” could be an overstatement and suggests to remove “great”

Response: We agree with the reviewer. Due to limited availability of the data on treatment administered to individual patient, we are not able to test the accuracy of the 12-Gene Algorithm on each type of chemotherapy. We agree with the reviewer’s suggestion and have removed “great” in the Abstract and the main text.

In general, it would be valuable for the reader to have an example of how many BC patients (e.g., out of a total of 100) candidates to (neo)adjuvant chemotherapy will have a correct prediction of their response to treatment based upon the sensitivity and specificity of the test, or, in other words, how many false negatives and false positives there would be. This simulation would provide clinicians with a better appreciation of the potential of this algorithm in clinical practice  

Response: We sincerely thank the comments made by the Reviewer! We have added two supplemental tables showing the number of positive, negative, false positive, and false negative patients predicted by the 12-Gene Algorithm in both the TCGA and the MSK cohorts as well as in the triple-negative and luminal A subgroups in the cohorts. We also cited the tables in the Results section accordingly.

MINOR OBSERVATIONS

Materials and Methods, Table 1 on page 4: TCGA cohort consists of 399 patients but when they are divided by BC type the sum is only 225 (42+155+14+14); the same for MSK cohort, instead of 807 patients the sum of different BC types is 597 (75+501+6+15)

Response: In the TCGA Cohort, 173 patients cannot be classified in any BC subtype because their HER2 status data was either missing or labeled “Equivocal”. Similarly in the MSK Cohort, 210 patients cannot be classified in any BC subtype. It’s our oversight that we didn’t include this information in Table 1. Now we have added the information in the revised Table 1 and the Materials and the Methods section. We greatly appreciate the reviewer for pointing out this omission!

Results page 6 line 237 please remove “Figure 1” on top of the figure, same for figure 2 (page 8), 3 (page 9), figure 4 on page 12 and figure 5 on page 14

Response: We have removed the figure labels as the reviewer suggested.

Results page 7 please insert an empty row between rows 257 and 258

Response: Revised according to the reviewer’s suggestion.

In many instances, results reported in tables and/or figures do not match with results in article body, or sometimes numbers appear to be rounded. This reviewer invites the authors to carefully double check each and every value reported in this article. Here below a non exhaustive list of such discrepancies:

Response: When citing the data from the tables and figures, we made some careless mistakes. Now we have carefully checked the whole manuscript and corrected all errors. We are very grateful to the reviewer for finding these errors and helping us improve the manuscript!

Results page 7 Table 2 when numbers do not always match those in Figure 2 on page 8: line 278, AUC of 0.89 but in Fig. 2 panel G AUC is 0.959; line 285, 69% appears to be 97% in Table 2

Response: The discrepancies were due to us putting the wrong ROC curves in Figure 2G and 2H. Now we have revised Figure 2 with the correct ROC curves and the AUCs match those cited in the Results. We have also corrected some careless errors in the tables.

Results figure 3 page 9: panel B “p=0.000” but on line 315 the manuscript reads “p<0.001”

Response: We have corrected the wrong p values cited in the Results.

Results table 3 HR for multivariate analysis in TNBC for cancer stage is 2.3 (0.36-14.0) with a p value of 0.382 but text on page 11 line 357 reads 3.8 (0.62-22.9) with a p value of 0.151

Response: We have corrected the wrong HR and p values in the table and the corresponding data cited in the Results.

Results table 3 univariate analysis for 12-gene algorithm in luminal A has a p value of p<0.0001 but text on page 11, line 365 reads p<0.001; the same for multivariate analysis (table vs line 369 on page 11)

Response: We have corrected the wrong p values cited in the Results.

Results table 3 HR for multivariate analysis cancer stage is 1.2 but text on page 11 line 371 reads 1.1 and the p values are 0.763 in table 3 and 0.835 in text on page 11, line 371

Response: We have corrected the wrong HR and p values in the table and the corresponding data cited in the Results.

Results page 11, lines 356 and 359 “tripe-negative” should be “triple-negative”

Response: We have corrected the wrong spelling.

Results page 12 line 390 0.97 should be 0.973

Response: We rounded all AUC values to two decimals following the general guidelines and have checked the whole manuscript to ensure all AUCs are consistently rounded up.

Results table 4 panel E and text page 13 line 408 authors should specify if numbers are rounded (0.876 vs 0.88, respectively)

Response: We have revised all incorrect AUC rounding.

Results page 13 lines 405 and 412 and figure 4D: please uniform to either 0.981 or 0.982

Response: We have revised all incorrect AUC rounding and the numbers cited in the manuscript.

References page 19 please remove 1. below ref nr. 30

Response: We have removed the number.

Response to Reviewer 2:

Comments and Suggestions for Authors

In manuscript, Jonsson et al utilized machine learning tools to develop gene-mutation-based algorithms, aiming at identifying reliable biomarkers to predict treatment response of breast cancer, especially triple-negative breast cancer. The authors used breast cancer TCGA cohort (n=399) as a training set and applied Random Forest machine learning to screen the algorithms of different combination of gene mutation profiles of patient samples. A novel 12-Gene Algorithm was developed, assessed, and validated in the MSK Cohort (n=807). Applying machine learning strategy in cancer therapy is a very exciting direction to explore. Jonsson et al here based on K-RAS associated gene-mutation profiles in breast cancer and show us the power of machine learning in prediction of treatment response of patients with subtypes of breast cancer, especially the deadly triple-negative breast cancer. The research overall is very meaningful.

However, given that the training set used here is breast cancer TCGA cohort (n=399) which is kind of limited, I suggest expanding training set and then further evaluate the algorithms based on more data sets.

Response: We sincerely thank the comments made by the Reviewer 2. We are intrigued by the Reviewer 2’s suggestions to expand the patient cohorts to further validate the algorithms. We plan to systematically conduct this large project by using the large patient cohorts from the muti-center partners. We hope to report our data from the expanded patient cohorts in the near future. 

Reviewer 2 Report

In manuscript, Jonsson et al utilized machine learning tools to develop gene-mutation-based algorithms, aiming at identifying reliable biomarkers to predict treatment response of breast cancer, especially triple-negative breast cancer. The authors used breast cancer TCGA cohort (n=399) as a training set and applied Random Forest machine learning to screen the algorithms of different combination of gene mutation profiles of patient samples. A novel 12-Gene Algorithm was developed, assessed, and validated in the MSK Cohort (n=807). Applying machine learning strategy in cancer therapy is a very exciting direction to explore. Jonsson et al here based on K-RAS associated gene-mutation profiles in breast cancer and show us the power of machine learning in prediction of treatment response of patients with subtypes of breast cancer, especially the deadly triple-negative breast cancer. The research overall is very meaningful. However, given that the training set used here is breast cancer TCGA cohort (n=399) which is kind of limited, I suggest expanding training set and then further evaluate the algorithms based on more data sets.

Author Response

To: Cancers, 

RE: Manuscript ID, cancers-1912723

Dear Editors and Reviewers,

We sincerely thank the Editors and Reviewers for providing valuable comments and suggestions in helping us improve the quality of our manuscript. We are especially encouraged by the facts that both Reviewers found our findings “interesting” and “very meaningful”. In the revised manuscript, we have addressed the important points by specifically following the instructions provided by the Editors and Reviewers. In addition, we have performed English writing revisions to improve the language and writing style of the manuscript, as suggested by the Editors and Reviewers. Your invaluable comments and suggestions have guided us to improve the quality of our manuscript so it can meet the high standards set by the journal. The revised text is marked in red. Below are our specific, point-by-point responses addressing the important suggestions of the Reviewers.

Response to Reviewer 1:

Reviewer 1: Comments and Suggestions for Authors

The manuscript reports on a retrospective evaluation of a 12-gene mutation score to predict treatment response in early breast cancer (BC). Treatment response is intended as progression-free survival after (neo)adjuvant chemotherapy.

The findings are interesting, and the authors acknowledge limitations of their work, in particular the retrospective nature of the analysis. Thus, this predictive algorithm could be tested prospectively as it potentially addresses an unmet medical need.

MAJOR OBSERVATIONS

The findings are retrospective in nature and based upon a small sample size, which becomes even smaller when the analysis is restricted to luminal A or TNBC patients. It is also unclear to this reviewer if the predictive value of the 12 gene mutation score is unaffected by the type of chemotherapy administered to individual patient (e.g., taxane vs platinum in TNBC). No information is available as to whether this patient population is homogeneous from this standpoint. It is this reviewer’s understanding that also early stage luminal A BC included in this analysis received chemotherapy.

Therefore, in abstract, page 2, line 47 this reviewer thinks that “great potential” could be an overstatement and suggests to remove “great”

Response: We agree with the reviewer. Due to limited availability of the data on treatment administered to individual patient, we are not able to test the accuracy of the 12-Gene Algorithm on each type of chemotherapy. We agree with the reviewer’s suggestion and have removed “great” in the Abstract and the main text.

In general, it would be valuable for the reader to have an example of how many BC patients (e.g., out of a total of 100) candidates to (neo)adjuvant chemotherapy will have a correct prediction of their response to treatment based upon the sensitivity and specificity of the test, or, in other words, how many false negatives and false positives there would be. This simulation would provide clinicians with a better appreciation of the potential of this algorithm in clinical practice  

Response: We sincerely thank the comments made by the Reviewer! We have added two supplemental tables showing the number of positive, negative, false positive, and false negative patients predicted by the 12-Gene Algorithm in both the TCGA and the MSK cohorts as well as in the triple-negative and luminal A subgroups in the cohorts. We also cited the tables in the Results section accordingly.

MINOR OBSERVATIONS

Materials and Methods, Table 1 on page 4: TCGA cohort consists of 399 patients but when they are divided by BC type the sum is only 225 (42+155+14+14); the same for MSK cohort, instead of 807 patients the sum of different BC types is 597 (75+501+6+15)

Response: In the TCGA Cohort, 173 patients cannot be classified in any BC subtype because their HER2 status data was either missing or labeled “Equivocal”. Similarly in the MSK Cohort, 210 patients cannot be classified in any BC subtype. It’s our oversight that we didn’t include this information in Table 1. Now we have added the information in the revised Table 1 and the Materials and the Methods section. We greatly appreciate the reviewer for pointing out this omission!

Results page 6 line 237 please remove “Figure 1” on top of the figure, same for figure 2 (page 8), 3 (page 9), figure 4 on page 12 and figure 5 on page 14

Response: We have removed the figure labels as the reviewer suggested.

Results page 7 please insert an empty row between rows 257 and 258

Response: Revised according to the reviewer’s suggestion.

In many instances, results reported in tables and/or figures do not match with results in article body, or sometimes numbers appear to be rounded. This reviewer invites the authors to carefully double check each and every value reported in this article. Here below a non exhaustive list of such discrepancies:

Response: When citing the data from the tables and figures, we made some careless mistakes. Now we have carefully checked the whole manuscript and corrected all errors. We are very grateful to the reviewer for finding these errors and helping us improve the manuscript!

Results page 7 Table 2 when numbers do not always match those in Figure 2 on page 8: line 278, AUC of 0.89 but in Fig. 2 panel G AUC is 0.959; line 285, 69% appears to be 97% in Table 2

Response: The discrepancies were due to us putting the wrong ROC curves in Figure 2G and 2H. Now we have revised Figure 2 with the correct ROC curves and the AUCs match those cited in the Results. We have also corrected some careless errors in the tables.

Results figure 3 page 9: panel B “p=0.000” but on line 315 the manuscript reads “p<0.001”

Response: We have corrected the wrong p values cited in the Results.

Results table 3 HR for multivariate analysis in TNBC for cancer stage is 2.3 (0.36-14.0) with a p value of 0.382 but text on page 11 line 357 reads 3.8 (0.62-22.9) with a p value of 0.151

Response: We have corrected the wrong HR and p values in the table and the corresponding data cited in the Results.

Results table 3 univariate analysis for 12-gene algorithm in luminal A has a p value of p<0.0001 but text on page 11, line 365 reads p<0.001; the same for multivariate analysis (table vs line 369 on page 11)

Response: We have corrected the wrong p values cited in the Results.

Results table 3 HR for multivariate analysis cancer stage is 1.2 but text on page 11 line 371 reads 1.1 and the p values are 0.763 in table 3 and 0.835 in text on page 11, line 371

Response: We have corrected the wrong HR and p values in the table and the corresponding data cited in the Results.

Results page 11, lines 356 and 359 “tripe-negative” should be “triple-negative”

Response: We have corrected the wrong spelling.

Results page 12 line 390 0.97 should be 0.973

Response: We rounded all AUC values to two decimals following the general guidelines and have checked the whole manuscript to ensure all AUCs are consistently rounded up.

Results table 4 panel E and text page 13 line 408 authors should specify if numbers are rounded (0.876 vs 0.88, respectively)

Response: We have revised all incorrect AUC rounding.

Results page 13 lines 405 and 412 and figure 4D: please uniform to either 0.981 or 0.982

Response: We have revised all incorrect AUC rounding and the numbers cited in the manuscript.

References page 19 please remove 1. below ref nr. 30

Response: We have removed the number.

Response to Reviewer 2:

Comments and Suggestions for Authors

In manuscript, Jonsson et al utilized machine learning tools to develop gene-mutation-based algorithms, aiming at identifying reliable biomarkers to predict treatment response of breast cancer, especially triple-negative breast cancer. The authors used breast cancer TCGA cohort (n=399) as a training set and applied Random Forest machine learning to screen the algorithms of different combination of gene mutation profiles of patient samples. A novel 12-Gene Algorithm was developed, assessed, and validated in the MSK Cohort (n=807). Applying machine learning strategy in cancer therapy is a very exciting direction to explore. Jonsson et al here based on K-RAS associated gene-mutation profiles in breast cancer and show us the power of machine learning in prediction of treatment response of patients with subtypes of breast cancer, especially the deadly triple-negative breast cancer. The research overall is very meaningful.

However, given that the training set used here is breast cancer TCGA cohort (n=399) which is kind of limited, I suggest expanding training set and then further evaluate the algorithms based on more data sets.

Response: We sincerely thank the comments made by the Reviewer 2. We are intrigued by the Reviewer 2’s suggestions to expand the patient cohorts to further validate the algorithms. We plan to systematically conduct this large project by using the large patient cohorts from the muti-center partners. We hope to report our data from the expanded patient cohorts in the near future. 

 Sincerely

Jenny Persson

Professor of Tumor Biology

Round 2

Reviewer 1 Report

The authors have improved the article and the accuracy of information provided

Just a few observations:

On line 162, “173” should be “174”

Line 243 figure legend is repeated

Line 297 “liminal” should be “luminal”